# The Percentage of Total and Regional Fat Is Negatively Correlated with Performance in Judo

**DOI:** 10.3390/sports11090168

**Published:** 2023-09-04

**Authors:** Izabela Aparecida dos Santos, Gabriel Felipe Arantes Bertochi, Wonder Passoni Higino, Marcelo Papoti, Enrico Fuini Puggina

**Affiliations:** 1Graduate Program in Rehabilitation and Functional Performance, Ribeirao Preto Medical School, University of São Paulo, Ribeirão Preto 14040-900, Brazil; izabelaeduca94@hotmail.com (I.A.d.S.); gabrielbertochi1@gmail.com (G.F.A.B.); 2Federal Institute of Education Science and Technology of the South of Minas Gerais, Muzambinho 37890-000, Brazil; wonderhigino@gmail.com; 3School of Physical Education and Sport of Ribeirão Preto, University of São Paulo, Ribeirão Preto 14040-900, Brazil; mpapoti@usp.br

**Keywords:** combat sports, martial arts, body composition, anthropometry, strength–power training

## Abstract

This study investigated the associations between total and regional body composition with performance in the special judo fitness test (SJFT), as well as strength and power tests (countermovement vertical jump—CMJ, squat jump—SJ, plyometric push-up—PPU, and force push-up—FPU). Twenty-three high-level judo athletes participated in this study. Initially, they underwent dual-energy X-ray absorptiometry, after which they performed the CMJ, SJ, PPU, and FPU tests. On another day, the SJFT was carried out. Correlations were tested using Pearson’s test. The performance in the SJFT was correlated with the total and arm %fat mass (r = −0.759), torso fat mass (r = −0.802), torso %fat mass (r = −0.822) and in the lower limb regions with the leg fat mass (r = −0.803) and leg %fat (r = −0.745). In the strength and power tests, there were also negative correlations observed between regional fat and performance. There was a negative correlation between the percentage of total fat and performance in the SJFT (r = −0.824), SJ (r = −0.750), CMJ (r = −0.742), PPU (r = −0.609), and FPU (r = −0.736). Fat, both total fat and regional fat in the arms, torso, and legs, is strongly correlated with a poor performance in the SJFT and poor strength and power.

## 1. Introduction

Judo is a complex combat sport that requires various abilities and characteristics to attain a high level in competitions [1]. Officially, judo athletes are categorized by weight and gender in competitions. Physically, judo is an intermittent and high-intensity sport with short recovery periods during combat. According to the rules, each combat lasts for 4 min, during which athletes strive to achieve the maximum score or finish the combat through an ippon or by immobilizing the opponent. To achieve success, judo athletes must demonstrate strength and muscular endurance to withstand the intermittent stimulus, grip and manipulate the opponent’s kimono, and execute throws [1,2]. However, studies indicate the relevance of parameters such as low levels of fat and high levels of free fat mass to improve performance in judo. This finding has drawn the attention of coaches regarding body composition [3,4].

In addition, body composition influences the maximum oxygen consumption (VO_2max_) and anaerobic power, which are two important physiological markers for judo. Aerobic metabolism is essential for effective recovery during the short intervals between strikes and combats in competition [1]. Additionally, the requirement to execute complex skills (techniques and tactics) within a short timeframe is facilitated by the anaerobic metabolism [5]. Thus, optimizing the body composition to enhance aerobic and anaerobic markers is crucial. Consequently, in accordance with the specificity of judo, the Special Judo Fitness Test (SJFT) has been extensively utilized among athletes in this discipline. The SJFT enables an analysis of the two aforementioned metabolic pathways (aerobic and anaerobic) through judo-specific movements, while also being cost effective and demonstrating proven efficiency [6,7].

Kratalli and Goudar (2012) observed a negative correlation between the fat percentage of Indian judo players and their performance in the SJFT (r = −0.690, *p* = 0.001). Therefore, a higher percentage of fat seems to be associated with poorer performance in body displacement activities, specifically in the SJFT [8]. However, the fat percentage obtained in Kratalli and Goudar’s (2012) study was determined using the skinfold method. It is important to note that the skinfold method is considered a doubly indirect method and is susceptible to errors when compared to gold standard methods like dual-energy X-ray absorptiometry (DXA). Moreover, the DXA method allows for the assessment of body composition by region, rather than just the overall body composition as measured by skinfolds [9].

Other tests routinely applied in judo, although not specific to technical skills, involve neuromuscular and strength performance, which are capabilities developed in judo. One example of such a test is the countermovement vertical jump (CMJ) [10]. This test is utilized in various sports and combat modalities and can serve as a useful tool for evaluating the neuromuscular power performance. Additionally, it can provide insights into the fatigue of athletes. Similar to the countermovement vertical jump (CMJ), the squat jump (SJ) is also widely popular and applicable for measuring lower limb power. A previous study established reference values for this test among high-level athletes across different sports. The study concluded that performance in this test is not influenced by sex and remains consistent within the practiced modality, enabling both men and women to be assessed within the same analysis [11].

When it comes to evaluating strength and power production in the upper limbs, variations of flexion exercises present an interesting strategy. In judo training routines, push-ups are commonly utilized. A previous study suggests that incorporating plyometric and traditional push-ups into training can enhance the upper limb strength. Although not specifically examined in the context of judo, plyometric and strength push-ups have been discussed in the existing literature. Given their prevalence in judo training, they can potentially serve as routine evaluation tests [12].

Performance measurements from these tests have been found to be correlated with the probability of winning a competition [10]. However, to the author’s knowledge, there is no evidence indicating a relationship between body composition and the performance of these tests.

It is worth noting that the current evidence focuses solely on total body composition. A previous study highlighted the significance of assessing regional body composition in sports performance, as it can provide insights into biomechanical aspects and power characteristics that differentiate performance levels in various sports. Furthermore, regional body composition assessments can serve as an indicator for identifying the risk of injuries. However, in the context of judo, no evidence exists regarding correlations between physical performance and regional body composition [13]. This makes the present study relevant in filling this gap. Therefore, the associations between total and regional body composition with performance in the Special Judo Fitness Test (SJFT), countermovement vertical jump (CMJ), squat jump (SJ), plyometric push-ups (PPU), and push-up force (FPU) were investigated among elite judo players.

## 2. Materials and Methods

### 2.1. Participants and Ethical Care 

Twenty-three high-level judo athletes (13 men (21 ± 3.8 years; 85 ± 16.7 kg; 176 ± 9.3 cm) and 10 women (21 ± 3.5 years; 63.5 ± 19.7 kg; 163 ± 10.1 cm)) participated in this study. 

All participants were taking part in the competitive period, with each athlete regularly participating in official national and international competitions. 

The protocols for this study were reviewed and approved by the local research ethics committee (CAAE: 12919719.3.0000.5659) and conducted according to the Declaration of Helsinki.

### 2.2. Procedures

Initially, in the morning, the judo athletes underwent a body composition evaluation using DXA. Following the body composition assessment, the athletes performed three maximum attempts of each test (CMJ, SJ, PPU, and FPU) with a three-minute rest period between tests.

After a twenty-four-hour period from the initial tests, the athletes returned to the laboratory for the execution of the Special Judo Fitness Test (SJFT). During the SJFT, athletes were individually evaluated and separated according to their weight categories.

### 2.3. Dual-Energy X-ray Absorptiometry-DXA

Body composition was assessed using the DXA method with the GE Lunar iDXA equipment (GE Health Care Lunar, Madison, WI, USA) and the Encore 2011 software, version 13.6 (Lunar iDXA, Madison, WI, EUA; software enCORE™ 2011 versão 13.6), for full body scanning. The variables considered in this study included the total fat percentage, the arm fat mass, the lean arm mass, the arm total mass, the arm fat mass percentage, the torso fat mass, the torso lean mass, the torso fat mass percentage, the leg fat mass, the leg lean mass, the leg total mass, and the leg fat mass percentage. To minimize measurement errors, the device was calibrated before each evaluation.

The judo athletes were positioned in dorsal decubitus and instructed to remain immobile, wearing minimal clothing and removing any metallic objects. The same technician conducted all assessments, and the procedures typically lasted approximately 15 min.

### 2.4. Assessment CMJ, SJ, PPU, and FPU

For the CMJ test, judo athletes were evaluated individually. The athletes were instructed to stand in a bipedal position with their hands placed on their hips. Upon receiving the signal, the athletes were required to flex their knees to approximately 90° before performing a maximal vertical jump, ensuring that their knees remained extended throughout the entire flight phase [14]. 

For the SJ test, the instructions were similar to the CMJ. The judo athletes started from a crouched position with their knees flexed at 90°. They maintained an upright torso, looking forward, with their hands on their hips, holding this position for three seconds. Upon the evaluator’s command, the participant was required to execute a powerful and rapid extension of the lower limbs [15]. It is important to note that if any of these specifications were not met, the jump was considered invalid.

For the PPU test, the judo athletes were given specific instructions. They were instructed to lower their chests, maintain a straight body position, and support most of their body weight with their hands (initial position). Upon receiving the signal, the athletes were required to push the ground as explosively as possible, fully extending their arms and losing contact between their hands and the ground, thus achieving maximum height during the ascending phase of flexion. When landing, the judo athletes were instructed to flex their arms upon re-establishing contact with the ground, aiming to reduce stress on the upper limb joints [16]. 

For the FPU (force push-up) test, the athlete was instructed to begin with their chest resting on the platform between their hands. Upon the evaluator’s signal, the athlete was required to perform a fast and explosive flexion movement, focusing solely on the concentric contraction [12]. 

In all of these tests, the judo players were given three attempts with a five-second interval between each attempt. The best performance, as determined by the maximum height reached, from the three attempts was considered for analysis in all of the described tests. A contact mat was used for the evaluation (CEFISE^®^, Nova Odessa, São Paulo. Model Jump System Duo, 600 × 300 × 8 mm).

### 2.5. SJFT

For the specific evaluation of judo athletes, the SJFT was utilized. According to Franchini et al. (2005), this test effectively assesses judo players at various skill levels [1]. After a specific warm-up consisting of judo throws, the athletes were grouped based on similar body mass and positioned within a 6 m area. The evaluated athlete, referred to as “Tori”, positioned themselves in the center of the area, while the other two athletes, “Uke A” and “Uke B”, were positioned on the left and right sides, respectively, each 3 m away from the center. 

The procedure consisted of three throwing sessions of 15 s (A), 30 s (B), and 30 s (C), with intervals of 10 s between them. During each session, the *Tori* threw the *Ukes* as often as possible, using the *ippon seoi nage* technique [17]. The total number of shots completed for the three sessions was used to measure performance. Heart rates were monitored immediately after the test, and 1 min after the end; the following formula was used to calculate the index:Index = (HRpost + HRpost1min)/number of throws

### 2.6. Statical Analysis

The Shapiro–Wilk test was used to assess the normality of the data. The degree of association between variables was examined using the Pearson correlation test between percentage of fat and test performance. A significance level of 5% was adopted for the analysis. GraphPad^®^ software (Prism 6.0, San Diego, CA, USA) was utilized for data analyses.

## 3. Results

Normality was accepted in all data presented. Correlations between upper body anthropometric parameters and SJFT performance are shown in Table 1.

Table 2 presents the correlation between lower body anthropometric parameters with SJFT performance.

Figure 1 shows the correlation between the total fat percentage and the number of SJFT throws. The value of r was −0.824, corresponding to *p* = 0.0001.

Table 3 presents the correlation between upper body anthropometric parameters and SJ, CMJ, FPU, and PPU performance.

Table 4 presents the correlation between lower body anthropometric parameters and SJ, CMJ, FPU, and PPU.

Figure 2A presents the correlation values between the SJ performance and the judo players’ total fat percentage. A value of r = −0.750 corresponds to *p* = 0.0001. Figure 2B shows the correlation between CMJ performance and the total fat percentage; the r value was −0.742, corresponding to *p* = 0.0001.

Figure 3A shows the correlation values between the FPU performance and the total fat percentage of the judo players. The value of r was −0.736, corresponding to *p* = 0.0001. Figure 3B shows the correlation between PPU performance and the total fat percentage; the r value was −0.609, corresponding to *p* = 0.002.

## 4. Discussion

In this study, potential associations between total and regional body composition and performance in elite judo players were investigated with SJFT, CMJ, SJ, PPU, and FPU. When analyzing the correlation between the upper body segment and SJFT, it was observed that there were negative correlations between the arm fat mass, arm %fat, torso fat mass, torso %fat, and all measured performances in this test (series A, B, C, total, and index). However, no correlations were found between the lean body mass and performance.

The SJFT has been explored from several aspects in the literature [18,19,20,21] due to the high level of correlation between its performance and judo success [21]. Body mass seems to be associated with the SJFT index. Thus, heavier judokas tend to have a worse performance compared to lighter ones. A previous study (assessed by skinfolds) revealed that the biceps measurement (mm) has a strong association with the performance in the SJFT, in which this measurement was able to explain 31% of the index of this test [22]. Therefore, arm composition is important for success in the SJFT, and our findings support this. We found that fat mass and the percentage of arm fat mass have a negative relationship with the SJFT performance. Thus, besides maintaining a lean mass in this region, reducing fat is also crucial, as it is a discriminating variable in the elite judo context [23]. 

Evaluating the arm composition in judo is rational. During combat, actions such as approaching or distancing from the opponent involve grabbing the kimono (collar or sleeve) and executing elbow extension to maintain distance, as well as flexion to approach the opponent [3]. Thus, a high level of lean mass and a lower fat mass in the arms can contribute to the strength of flexion and extension actions. Strength has already been demonstrated as proportional to the size of the muscle [24]; therefore, the leaner mass, the bigger the muscles, the more strength being produced, and the more advantages in judo combat [21]. Additionally, the opposite scenario would be that an increase in fat in this region compromises strength and worsens performance. Our findings reinforce these assertions, as strong negative correlations were found between all components of the SJFT (series A, B, C, total, and index) and the arm fat mass as well as arm %fat.

The torso is a body region frequently used in judo techniques. The strength development in the torso is associated with specific performance parameters in judo players [25]. Our findings indicate that fat accumulation in the trunk region has a negative impact on performance in the SJFT, as well as in the other evaluated tests (CMJ, SJ, FPU, and PPU). Furthermore, these data draw attention to the health of athletes, as the distribution of fat is an aspect that should be considered, particularly in the torso region, as it plays a crucial role in the occurrence of metabolic disorders [26].

Negative correlations were observed between leg fat mass, leg %fat, and the performance of all the tests involving this region in this study (SJ, CMJ, and SJFT), as well as tests that do not heavily rely on leg involvement (PPU and FPU). These findings can be explained by the fact that fat mass is considered an additional non-functional weight in sports. Therefore, carrying this “extra weight” leads to decreased jumping heights (e.g., SJ and CMJ) and an impaired locomotion capacity (e.g., SJFT) [27].

Overall, the total fat percentage had a negative correlation (r = −0.824, *p* = 0.0001) with the overall SJFT performance, aligning with a previous study [21]. Observing the performance classification in this test, the male judo players had an average number of throws of 26.3, that is, they are classified between “regular” and “good” according to a study by Stercowicz-Przybycień et al. (2019) [28]. On the other hand, women performed about 25.1 throws, which is classified as “regular” according to Agostinho et al. (2018) [29]. The SJFT is correlated with parameters of anaerobic and aerobic capacity; a previous study carried out with soccer players revealed that there are negative correlations between fat mass and the performance of these parameters, the anaerobic capacity evaluated by sprints and jumps tests, and the aerobic capacity by a test measuring the VO_2max_ [30].

The percentage of total fat showed negative correlations with tests of strength and power characteristics (SJ: r = −0.750, *p* = 0.0001; CMJ: r = −0.752, *p* = 0.0001; FPU: r = −0.736, *p* = 0.0001; PPU: r = −0.609, *p* = 0.002). This can be explained by the negative relationship between fat mass and muscle performance. There is evidence that adiposity can have a detrimental effect on muscle activation. In summary, an increased fat mass primarily reduces the activation of muscle agonists [31].

Some limitations need to be addressed. There was no separation of the judo players by gender, as doing so would significantly reduce the number of participants. Additionally, there was no separation based on weight categories. However, despite these limitations, this study is the first to investigate the relationship between localized fat and performance in judo tests, both specific and non-specific. Further research should explore training strategies targeted at reducing overall and regional fat, while also considering the specificity of the sport.

## 5. Conclusions

Fat accumulation in the arms, torso, and legs is strongly correlated with a poor performance in the SJFT. Additionally, both the regional and total fat are negatively correlated with strength and power.

Reducing fat and maintaining lean mass contribute to improved test performances, leading to better manifestation of motor skills and potentially enhancing sport performance. However, it is important to note that competition outcomes in official fights are influenced by multiple factors, and the relationship between body composition and performance is not a simple cause-and-effect relationship in such contexts. Nevertheless, this association holds true for performance in the tests conducted in this study.

Therefore, judo coaches and physical trainers should consider an anthropometric analysis that includes variables beyond the commonly used ones such as body mass index and total fat percentage. Alongside preserving lean mass, decreasing the percentage of fat (both total and regional) in athletes is crucial for achieving a satisfactory performance. Furthermore, tests like PPU and FPU should be further explored for evaluations of judo players.

## Figures and Tables

**Figure 1 sports-11-00168-f001:**
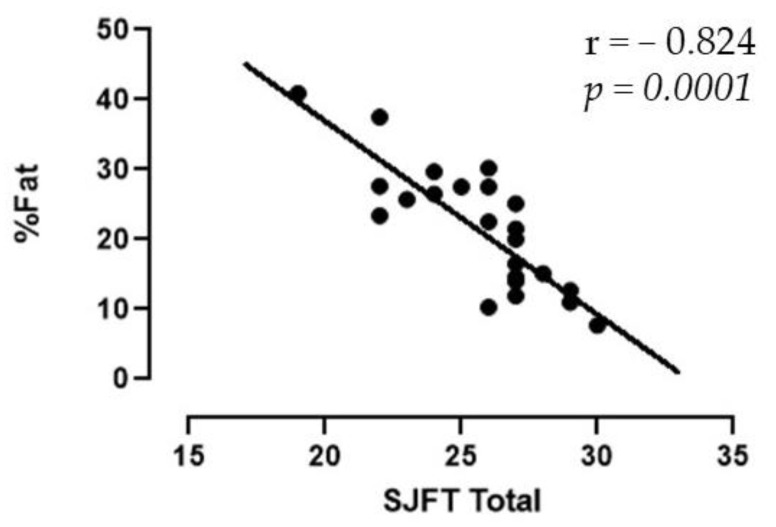
Correlation between the percentage of total fat and total number of throws in the SJFT.

**Figure 2 sports-11-00168-f002:**
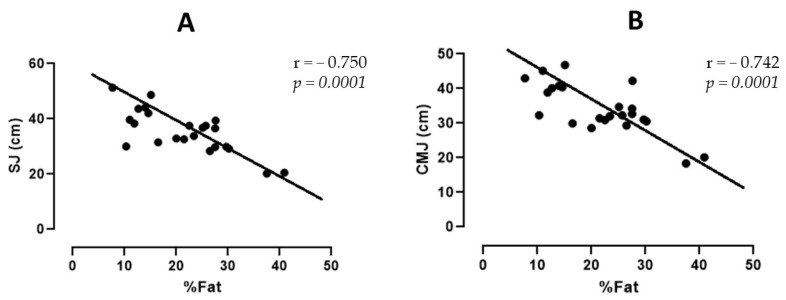
(**A**) Correlation between the performance of SJ with the total fat percentage. (**B**) Correlation between the performance of CMJ with the total fat percentage. SJ = squat jump; CMJ = countermovement vertical jump; cm: centimeters.

**Figure 3 sports-11-00168-f003:**
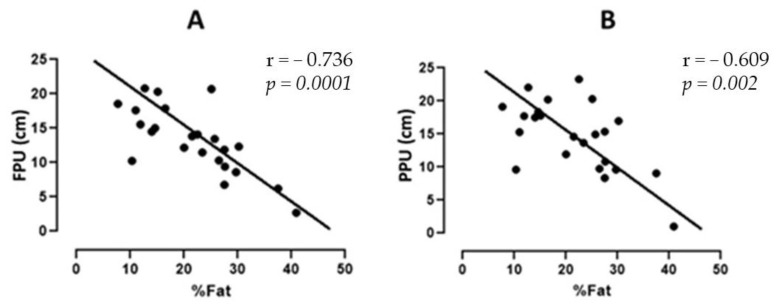
(**A**) Correlation between the performance of FPU with total fat percentage. (**B**) Correlation between the performance of PPU with total fat percentage. FPU = push-up force; PPU = plyometric push-ups; cm: centimeters.

**Table 1 sports-11-00168-t001:** Anthropometric parameters of the upper limbs and performance during the SJFT.

	Arm Fat Mass	Arm Lean Mass	Arm Total Mass	Arm %Fat Mass	Torso Fat Mass	Torso Lean Mass	Torso %Fat
SJFT Serie A	r = −0.538 *	r = 0.170	r = −0.020	r = −0.469 *	r = −0.582 *	r = 0.141	r = −0.542 *
SJFT Serie B	r = −0.715 *	r = 0.164	r = −0.089	r = −0.701 *	r = −0.764 *	r = 0.050	r = −0.774 *
SJFT Serie C	r = −0.770 *	r = 0.272	r = −0.004	r = −0.805 *	r = −0.786 *	r = 0.196	r = −0.841 *
SJFT total	r = −0.765 *	r = 0.234	r = −0.038	r = −0.759 *	r = −0.802 *	r = 0.149	r = −0.822 *
Index	r = −0.479 *	r = −0.371	r = −0.181	r = −0.602 *	r = 0.559	r = −0.322	r = −0.666 *

SJFT = Special Judo Fitness Test. Index = (HRpost + HRpost1min)/number of throws; % = percentage; * = *p* < 0.05.

**Table 2 sports-11-00168-t002:** Anthropometric parameters of the lower limbs and performance during the SJFT.

	Leg Fat Mass	Leg Lean Mass	Leg Total Mass	Leg %Fat
SJFT Serie A	r = −0.480 *	r = −0.037	r = −0.299	r = −0.414 *
SJFT Serie B	r = −0.790 *	r = −0.145	r = −0.547 *	r = −0.713 *
SJFT Serie C	r = −0.823 *	r = −0.075	r = −0.518 *	r = −0.798 *
SJFT total	r = −0.803 *	r = −0.097	r = −0.522 *	r = −0.745 *
Index	r = 0.533 *	r = −0.112	r = 0.225	r = 0.612 *

SJFT = Special Judo Fitness Test. Index = (HRpost + HRpost1min)/number of throws; % = percentage; * = *p* < 0.05.

**Table 3 sports-11-00168-t003:** Anthropometrics parameters of the upper limbs and performance in tests.

	Arm Fat Mass	Arm Lean Mass	Arm Total Mass	Arm %Fat	Torso Fat Mass	Torso Lean Mass	Torso %Fat
SJ	r = −0.573 *	r = 0.637 *	r = 0.391	r = −0.808 *	r = −0.472 *	r = 0.582 *	r = −0.617 *
CMJ	r = −0.633 *	r = 0.492 *	r = 0.242	r = −0.788 *	r = −0.529 *	r = 0.413 *	r = −0.624 *
FPU	r = −0.605 *	r = 0.483 *	r = 0.237	r = −0.768 *	r = −0.540 *	r = 0.391	r = −0.650 *
PPU	r = −0.499 *	r = 0.412	r = 0.206	r = −0.625 *	r = −0.451 *	r = 0.290	r = −0.494 *

SJ = squat jump; CMJ = countermovement vertical jump; FPU = push-up force; PPU = plyometric push-ups; * = *p* < 0.05.

**Table 4 sports-11-00168-t004:** Anthropometrics parameters of the lower limbs and performance in tests.

	Leg Fat Mass	Leg Lean Mass	Leg Total Mass	Leg %Fat
SJ	r = −0.655 *	r = 0.394	r = −0.103	r = −0.814 *
CMJ	r = −0.687 *	r = 0.249	r = −0.219	r = −0.796 *
FPU	r = −0.700 *	r = 0.140	r = −0.299	r = −0.742 *
PPU	r = −0.677 *	r = 0.102	r = −0.314	r = −0.655 *

SJ = squat jump; CMJ = countermovement vertical jump; FPU = push-up force; PPU = plyometric push-ups; * = *p* < 0.05.

## Data Availability

The data are not publicly available due to progress of other studies.

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
