# Peer review of "The Percentage of Total and Regional Fat Is Negatively Correlated with Performance in Judo"

_sports, 2023, doi:10.3390/sports11090168_

Round 1
Reviewer 1 Report
1.- The body composition and the performance tests were made during a competitive or training period?
2.- I consider that it is important to separate men from women due to differences in body composition per se
3.- You conclude that fat in the arms, torso, and legs are strongly correlated with poor performance in the specific judo test and the percentage of total fat...... these findings influenced your participants to win or lose a competition?
Author Response
Thank you very much for your comments on the manuscript. Attached are the responses and changes made.

Reviewer 2 Report
Dear Authors,
The article needs to be improved. I present some suggestions/comments:
Major issues.
What is new in this article? As the authors indicated It was already known that “low levels of fat and high levels of free fat mass to improve performance in judo, which has 36 drawn the attention of coaches around body composition [3,4].” So why this article should be published? The article needs to be well justified in order to be published.
Minors.
The document should be written better:
· ..(SJFT) and strength and power tests.. To delete the first “and”
· Why “.The same happened with the PPU (r = - 0.609)…” is separated from the other physical tests?
· “to hold the opponent's kimono and move and throw it” you mean him or her
· “intervals and trials during and between trial combats”
· ….
Some contradiction: In the abstract, why DXA was performed after the physical tests? It used to be performed before any tests, “absorptiometry after athletes performed the tests: CMJ, SJ, PPU, and FPU” However in the Materials and Methods, the DXA is done before de tests.
The correlations with total fat are included in the abstract, why not the correlations with regional fat?
To correct the units or numbers: example “176 ± 9.3 80 m”
Which was the criteria to consider “high-level judo athletes”?
Do not repeat. The meaning of DXA was explain before line 95, why it is indicated again in line 95?
To be uniform:
· Why the abbreviation is included in some headings “2.5. Special Judo Fitness Test – SJFT” and not in others?
· Why the abbreviation SJFT is not used in the conclusion when PPU, FPU are used it?
· …..
All tables and figures must be understood by readers without checking the article, so the meaning of all abbreviation must be included in all of them.
Conclusion. “Fat in the arms, torso, and legs are strongly correlated with poor performance in the specific judo test and the percentage of total fat” it is obvious that total fat should correlate with regional fat.
Best Regards
I already indicate that the article should be written better (contents and English)
Author Response

(The authors gave the same response as above.)
